# Quality of Life and Mental Health Status of Japanese Older People Living in Chiang Mai, Thailand

**DOI:** 10.3390/geriatrics6020035

**Published:** 2021-03-30

**Authors:** Takeshi Yoda, Bumnet Saengrut, Benjamas Suksatit, Kanae Kanda, Hiromi Suzuki, Rujee Rattanasathien, Rujirat Pudwan, Hironobu Katsuyama

**Affiliations:** 1Department of Health and Sports Science, Kawasaki University of Medical Welfare, Kurashiki 701-0193, Japan; 2Department of Public Health, Kawasaki Medical School, Kurashiki 701-0192, Japan; katsu@med.kawasaki-m.ac.jp; 3Nursing Service Department, Maharaj Nakorn Chiang Mai Hospital, Faculty of Medicine, Chiang Mai University, Chiang Mai 50200, Thailand; xbumnet@gmail.com (B.S.); eejur@hotmail.com (R.R.); rujirat.pu@gmail.com (R.P.); 4Faculty of Nursing, Chiang Mai University, Chiang Mai 50200, Thailand; benjamas.s@cmu.ac.th; 5International College of Digital Innovation, Chiang Mai University, Chiang Mai 50200, Thailand; 6Department of Public Health, Faculty of Medicine, Kagawa University, Takamatsu 761-0793, Japan; oda@med.kagawa-u.ac.jp; 7Department of Hygiene, Faculty of Medicine, Kagawa University, Takamatsu 761-0793, Japan; tanzukimama@yahoo.co.jp

**Keywords:** aged, Japan, mental health, quality of life, surveys and questionnaires

## Abstract

This study aimed to establish the quality of life and mental health status among older Japanese people living in Chiang Mai, Thailand. We conducted a questionnaire survey among Japanese retired people aged 50 years or over who had been living in Thailand. The questionnaire covered socio-demographic variables including health status and ability to communicate in Thai. We measured mental health status using the Japanese version of the General Health Questionnaire-28 (GHQ-28) and quality of life using the Japanese version of EuroQOL-5D-3L. We explored the factors associated with poor mental health and quality of life using logistic regression analysis. In total, 96 (89.7%)participants provided complete responses. Overall, quality of life was generally good, although those with one or more chronic diseases reported significantly lower quality of life. Having one or more chronic diseases and being aged 70–79 were significantly associated with poorer mental health. In total, 21 (21.8%) respondents had a possible neurosis, which was defined as a total GHQ-28 score of more than 6. The logistic regression analysis showed a significant association between possible neurosis and the presence of chronic diseases (adjusted odds ratio: 11.7 1). Quality of life among older Japanese people living in Chiang Mai was generally good, but there was a high level of possible neurosis, especially among those with one or more chronic diseases.

## 1. Introduction

International retirement migration is a term used in the 1980s and early 1990s to describe the movement of older people between countries, often for tourist reasons rather than following standard migration models [1]. International retirement migration is increasingly popular among Japanese older people, as it is encouraged by the Japanese Ministry of International Trade and Industry, which launched the Silver Columbia Project to encourage older people to go on long-stay trips or move abroad [2]. The term “long-stay” is a commercial term and a trademark of the Long Stay Foundation, which was registered in 1992. The Foundation defined “long-stay” as staying overseas for a relatively long time but with the expectation of returning to Japan. Therefore, it does not involve migration or permanent residence in a foreign country. Long-stayers tend to own or rent a property rather than staying in a hotel. Long-staying is voluntary and usually to make use of leisure time, and it aims to seek “life” (ordinary experience) rather than “travel” (package tours) [3]. The source of income used to pay for it should be from Japan (for example, pension, interest from a bank account, dividends, or remittances), and income from working at the destination should not be necessary [3]. Long-stay tourism has developed as a form of Japanese international retirement migration. Hawaii, Europe, Australia, and New Zealand are considered popular but expensive destinations, and Southeast Asian countries are chosen mainly for economic reasons, although many Japanese people do find Asian culture attractive [4]. Thailand is one of the most popular destinations for Japanese older people [5]. The registration data for overseas residents in Thailand from 2017 show that the number of Japanese expatriates doubled in the previous decade, and it was around 64,000 [6]. The exact number of long-stay retirees in Thailand is unknown because of the variety of types of stays and visas [7,8], but Immigration Bureau statistics suggest that in 2014, at least 3000 Japanese people stayed in Thailand for over a year using retirement visas, which was up from 1400 in 2007 [9]. Many Japanese retirees were healthy and wealthy when they moved to Thailand, so could enjoy hobbies and experiencing a different culture during a long stay. However, those who became sick identified the language barrier as an obstacle when visiting hospitals [2]. Japanese older people living in Thailand seem to be happy, but their quality of life and mental health status are unknown. Therefore, the purpose of this study was to establish their quality of life and mental health status using a questionnaire survey, through cooperation with clubs for Japanese older people in Chiang Mai, Thailand.

## 2. Materials and Methods

### 2.1. Target Population

We conducted a questionnaire survey in 2016. The target population was long-stay Japanese older people in Chiang Mai province, Thailand, defined as those (1) at least 60 years old, (2) retired from work in Japan with a desire to stay in Thailand, and (3) living in Thailand for at least 3 months [2]. Chiang Mai, in the northern part of Thailand, was chosen as the study site because the city of Chiang Mai is a popular location for Japanese long-stayers [10], and the Chiang Mai province is one of five pilot provinces for long-stay tourism in Thailand [2]. The province has many attractions for long-stay tourists, including easy accessibility, a full range of accommodation options, hospital services, travel services, food and beverages, touring programs, and health and spa services [10]. There are more medical translators in Chiang Mai province than in any other area in Thailand except for the capital, Bangkok [2]. Participants were recruited from two clubs for Japanese older people in Chiang Mai, the Chiang Mai Long-stay Club for Japanese People and the Chiang Mai Expatriates Club for Japanese People. These clubs hold monthly meetings, so we visited each club for its meeting in February, 2016. At the meeting, we explained this study’s purpose and research details, and after obtaining agreement, we provided a self-reported questionnaire for each person.

### 2.2. Contents of the Questionnaire

The questionnaire was in three parts. The first covered socio-demographic variables such as age, gender, marital status, number of people living together, educational level, annual income, region of residence, length of time living in Thailand, annual number of return trips to Japan, language skills in Thai, present status of chronic diseases, frequency of visits to clinics/hospitals, smoking status, alcohol drinking habits, weekly exercise, and hobbies. The second part measured mental health status using the General Health Questionnaire-28 items Japanese version (GHQ-28) [11]. This is a self-administered tool originally developed to screen for nonpsychotic psychiatric illnesses. It is used in both clinical settings and epidemiologic studies to investigate the mental health of populations. The Japanese version used in this study contains 28 items in four categories: somatic symptoms, anxiety and insomnia, social dysfunction, and depression. Using the GHQ method (two-step method: 0-0-1-1 points), scores were calculated with a maximum of 28 points [12,13,14,15]. The third section measured quality of life using the Japanese version of EuroQOL-5D-3L (EQ-5D) [16]. The EQ-5D measures generic preference-based health status and provides utility scores. It is well-known and one of the most commonly used evaluation tools for quality of life worldwide. EQ-5D-3 Level (3L) is self-evaluated by respondents across three levels (no problems, some problems, and extreme problems), and converted utility scores are used in a country-specific tariff [17,18,19]. In this study, the answers were converted to utility scores using the Japanese tariff [20].

### 2.3. Data Analysis

Descriptive statistics were used to evaluate mental health status and quality of life by age groups, marital status, and chronic diseases. We used a cut-off of 5/6 on GHQ-28 to indicate potential disparities in depressive symptoms, drawing on previous research [21]. We explored the factors associated with potential disparities in depressive symptoms using logistic regression analysis. Independent variables were age groups, gender, marital status, and chronic diseases. The 95% confidence interval (CI) was used to determine the statistical significance of the odds ratio. JMP Pro 14.1.0 (SAS Institute Inc. Cary, NC, USA) was used for all analyses.

### 2.4. Ethical Concerns

The study was approved by the ethical committee of Kagawa University (Approval number: H27-126). Implied consent was used, rather than formal written consent, to assure anonymity of participants (that is, we considered that a completed and returned questionnaire showed consent for participation in the survey). This implied consent principle was explained to the participants beforehand. A letter of approval was also obtained from representatives of each Japanese club.

## 3. Results

In total, 107 Japanese older people agree to participate, of whom 96 (89.7%) provided complete responses. The mean age of respondents was 69.6 ± 5.9 years, 88% went back to Japan at least once a year (mean = 2.1 times/year), and 67 people had one or more chronic diseases. Details of socio-economic status are shown in Table 1. One-way ANOVA results for quality of life and GHQ scores are shown in Table 2. Chronic disease was significantly associated with seriousness of GHQ score. In addition, people with chronic diseases were significantly lower of EQ-5D tariff than the other.

We defined “possible neurosis” as a total GHQ-28 score of more than 6 points, drawing on a previous Japanese study [21], and 21 (21.8%) respondents fell into this category.

Table 3 shows the results of the logistic regression analysis for possible neurosis and other socio-demographic variables. Model 1 was adjusted for gender, age group, and the presence of chronic diseases. There was a significant association between possible neurosis and having one or more chronic diseases (adjusted odds ratio: 10.88, 95% CI: 1.29–91.74). Model 2 was adjusted for the variables of model 1 plus marital status and period of residence in Chiang Mai. This model also showed a significant association between possible neurosis and presence of chronic diseases (adjusted odds ratio: 11.71, 95% CI: 1.35–101.45). No significant association was found for possible neurosis and other socio-demographic variables.

## 4. Discussion

The purpose of this study was to establish the quality of life and mental health status of Japanese retirees living in Chiang Mai, Thailand. The key findings were the average EQ-5D scores, the average GHQ-28 scores, and detection of a high level of possible neuroses, plus a potential link between possible neurosis and the presence of chronic diseases.

Fujikawa et al. used EQ-5D to assess quality of life among Japanese older people living in a rural area of Japan [22]. They found average values of 0.868 for those aged 60 to 70 years, 0.780 for those aged 70 to 80 years, and 0.684 for those aged 80 and over. The figures in our study were higher for every age group. It is hard to compare these results because the background of the respondents was very different, but our findings suggest that older people living in Chiang Mai may feel more comfortable than those living in Japan. Ono suggested that the majority of Japanese long-stay retirees in Malaysia are there because of the leisure activities [3]. We think this may also be true of other Southeast Asian countries including Thailand.

However, the GHQ-28 results suggest that not all Japanese older people are in good health. People with chronic diseases showed significantly higher overall GHQ average score and also average scores for sections on somatic symptoms, anxiety symptoms, and social dysfunction. In addition, their EQ-5D tariffs were significantly lower than those for people without chronic diseases. Generally speaking, chronic diseases have an enormous effect on the psychiatric condition and their quality of life of those affected [23,24,25,26]. Our study results show that most Japanese older people cannot read and write Thai, and four-fifths cannot speak Thai, even though half of them have lived in Chiang Mai for more than 5 years (see Table 1). Moreover, the conversation skills were not associated for GHQ score nor EQ-5d tariff (see Table 2). This means that they may find it hard to go to Thai clinics to seek healthcare, and this may make them feel anxious.

Our logistic regression analysis strongly supported the relationship between possible neurosis and chronic diseases. Even when we adjusted for age, gender, marital status, and length of stay in Thailand, the odds ratio of an association with possible neurosis was still significantly higher for having one or more chronic illnesses than none. One of the most frequently mentioned factors associated with QOL in old age is supposed to be health [27,28]. To keep their health is quite important not only for older people living outside of their native country but also every older person. However, once they have chronic diseases, they face many barriers such as language, difference of culture, and discrimination [29,30]. Since they are a minority compared with native older people, a supportive system is needed from their country.

Our study had some limitations. First, the study sample was not randomized. The study population itself is small, and it is not feasible to distribute questionnaires to people’s homes, so this approach was considered a realistic way to obtain a suitable study sample. However, the sample size was relatively small, so the confidence interval was wide, and it is difficult to understand some of the results statistically. Third, the study was cross-sectional, so cannot explain any causality. Fourth, the study design was quantitative, and some personal matters were omitted. A qualitative study would be recommended for in-depth exploration in the future.

Despite these limitations, our study showed the real situations of Japanese older people after retirement migration from Japan to Thailand. At first glance, they seem to have a higher quality of life than people of the same age living in Japan. However, some of them, especially those with chronic diseases, are worried about their health. As a result, many may have anxieties and possible neurosis. It is important to support those with chronic diseases to live a better life and remain healthy, and this is likely to apply to many older people living abroad, not just Japanese people living in Chiang Mai.

## Figures and Tables

**Table 1 geriatrics-06-00035-t001:** Socio-demographic status of Japanese older people living in Chiang Mai.

Variables		N	%
Gender	Male	70	72.9
	Female	26	27.1
Age group (years)	60–64	21	21.9
	65–69	29	30.2
	70–74	28	29.1
	75–79	9	9.4
	80 or over	9	9.4
Chronic diseases	None	29	30.2
	One or more	67	69.8
Marital status	Single	22	22.9
	Married	74	77.1
Ability to read/write Thai	Poor	70	72.9
	Fair	16	16.7
	Average/Good/Excellent	10	10.4
Ability to speak Thai	Poor	35	36.5
	Fair	39	40.6
	Average/Good/Excellent	22	22.9
Period of residence	<5 years	48	50.0
	More than 5 years	48	50.0

**Table 2 geriatrics-06-00035-t002:** Scores for each item in the GHQ-28 and EQ-5D by age, chronic diseases, and mental health status.

	GHQ-28										EQ-5D	
	Somatic Symptoms	*p*	Anxiety Symptoms	*p*	Social Dysfunction	*p*	Severe Depression	*p*	Total Score	*p*	Tariff	*p*
**Age group**												
60–64years	0.904 ± 1.578	0.66	1.047 ± 0.973	0.12	0.380 ± 0.589	0.16	0.047 ± 0.218	<0.01 *	2.380 ± 2.246	0.07	0.961 ± 0.096	0.05
65–69 years	1.207 ± 1.448		1.000 ± 1.164		0.482 ± 0.870		0		2.689 ± 2.941		0.952 ± 0.114	
70–74 years	1.535 ± 1.574		2.035 ± 1.971		0.928 ± 1.358		0.392 ± 0.875		4.892 ± 4.524		0.868 ± 0.161	
75–79 years	1.444 ± 2.006		1.555 ± 2.242		1.222 ± 1.481		0.777 ± 1.394		5.000 ± 5.220		0.910 ± 0.139	
80 years or over	0.888 ± 1.964		1.222 ± 2.048		0.555 ± 1.013		0		2.666 ± 4.743		0.850 ± 0.199	
**Chronic diseases**												
None	0.400 ± 0.621	<0.01 *	0.666 ± 0.802	<0.01 *	0.233 ± 0.504	<0.01*	0.300 ± 0.876	0.43	1.600 ± 1.631	<0.01 *	0.980 ± 0.075	<0.01 *
One or more	1.656 ± 1.771		1.746 ± 1.795		0.850 ± 1.209		0.179 ± 0.601		4.432 ± 4.214		0.890 ± 0.154	
**Marital status**												
Single	1.500 ± 1.81	0.44	1.363 ± 1.328	0.87	0.590 ± 1.053	0.73	0.272 ± 0.767	0.66	3.727 ± 3.534	0.81	0.938 ± 0.125	0.45
Married	1.200 ± 1.559		1.426 ± 1.717		0.680 ± 1.092		0.200 ± 0.677		3.506 ± 3.946		0.912 ± 0.145	
**Ability to speak Thai**												
Poor	1.314 ± 1.529	0.84	1.342 ± 1.392	0.81	0.800 ± 1.255	0.64	0.285 ± 0.859	0.62	3.742 ± 0.525	0.91	0.906 ± 0.148	0.78
Fair	1.333 ± 1.721		1.333 ± 1.675		0.615 ± 1.041		0.153 ± 0.586		3.435 ± 3.912		0.929 ± 0.127	
Average/Good/Excellent	1.090 ± 1.659		1.590 ± 1.943		0.545 ± 0.857		0.136 ± 0.467		3.363 ± 4.381		0.914 ± 00.156	

Probability (*p*) was calculated by ANOVA for age groups, and t-test for others. * *p* < 0.05.

**Table 3 geriatrics-06-00035-t003:** Adjusted odds ratio (AOR) and 95% confidence interval (CI) of logistic regression analysis for possible neurosis ^†^.

		Model 1 *		Model 2 **	
Variables		AOR	95% CI	AOR	95% CI
Gender	Male	1.00		1.00	
	Female	0.64	0.14–2.81	0.77	0.16–3.69
Age group (years)	60–64	1.00		1.00	
	65–69	1.11	0.21–5.73	1.04	0.19–5.68
	70–74	2.76	0.59–12.88	2.37	0.46–12.18
	75–79	3.24	0.44–23.62	3.42	0.43–26.74
	80 or more	0.64	0.05–7.95	0.61	0.04–8.08
Chronic illness	None	1.00		1.00	
	One or more	10.88	1.29–91.74	11.71	1.35–101.45
Marital status	Single	-		1.00	
	Married	-		0.58	0.16–2.13
Length of stay in Thailand	<5 years	-		1.00	
	5 years or more	-		0.59	0.18–1.86

^†^: Possible neurosis was defined as a total score of more than 6 points for GHQ-28. *: Model 1 was adjusted for gender, age group, and chronic diseases. **: Model 2 was adjusted for gender, age group, chronic illness, marital status, and living period. Model evaluation: Model 1: Akaike′s information criterion (AIC) 98.10, R2 0.155, Model 2: AIC 101.55, R2 0.168. AOR: Adjusted odds ratio, CI: Confidence interval.

## Data Availability

The data presented in this study are available on request from the corresponding author (T.Y.). The data are not publicly available due to privacy concerns.

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
