# Peer review of "Quality of Life and Mental Health Status of Japanese Older People Living in Chiang Mai, Thailand"

_geriatrics, 2021, doi:10.3390/geriatrics6020035_

Round 1

Reviewer 1 Report

Thanks to authors for valuable study regarding quality of life of Japanese older people living abroad! But I have some comments regarding terminology used in manuscript and results/ discussion.

  1. In introduction authors gave concise explanation on aspects of international retirement migration and long-stay overseas. I think that the concept "long-stay Japanese older people living in Chiang Mai province, Thailand" (stated to describe target population) should be used more consequently through the manuscript, including title. In my opinion, term "living" has also other meaning than this long-stay related to international retirement migration.
  2. In Table 1, the period of residence is presented very detailed categories. As this information later is not utilised in such details, I would suggest to present only two categories - less than 5 years and more that 5 years (such dichotomous variable is used for logistic regression analysis).
  3. In my opinion the discussion could be improved in the manuscript. Despite some limited comparison of results with the findings from other studies and listing few limitations, I would recommend to strenghten discussion to show value of the study in terms of problematization older people living out of native country situation or implications of results in order to support their situation. In fact, I got a lot of questions for myself about participants real life situation and,  perhaps, qualitative study would be recommended for in-depth exploration in future.  

Author Response

Thank you very much to read very carefully for our manuscript. We changed some parts and re-analysed according to your suggestions. We have provided point-by-point responses to your comments.

  • In introduction authors gave concise explanation on aspects of international retirement migration and long-stay overseas. I think that the concept "long-stay Japanese older people living in Chiang Mai province, Thailand" (stated to describe target population) should be used more consequently through the manuscript, including title. In my opinion, term "living" has also other meaning than this long-stay related to international retirement migration.

Thanks to your comments.  We emphasized about concept of "long-stay" Japanese older people in discussion section (page 7, line 217-223). 

  • In Table 1, the period of residence is presented very detailed categories. As this information later is not utilised in such details, I would suggest to present only two categories - less than 5 years and more that 5 years (such dichotomous variable is used for logistic regression analysis).

Thank you very much for your suggestion. We changed "the period of residence" category to 2 items according to your opinion (page 4, the bottom of table 1).

  • In my opinion the discussion could be improved in the manuscript. Despite some limited comparison of results with the findings from other studies and listing few limitations, I would recommend to strenghten discussion to show value of the study in terms of problematization older people living out of native country situation or implications of results in order to support their situation. In fact, I got a lot of questions for myself about participants real life situation and,  perhaps, qualitative study would be recommended for in-depth exploration in future.  

Thank you very much your opinion. We have added some opinions in the discussion section (page 7, line 217-223) and also added for the limitation section (page 7, line 229-231).

Reviewer 2 Report

Excellent paper.

Author Response

Thank you very much for reading our manuscript. 

Reviewer 3 Report

The idea to study the group of long-stayers with regard to selected parameters of their wellbeing while abroad is interesting and has the potential for meaningful analyses.

In the context of the reviewed study, the selection of analyzed parameters needs justification. Why is, for example, no parameter of functional capacity included? The conclusions in the manuscript are weak and, as such do not deliver a valid clue.

Furthermore, the choice of analyzed age cohorts raises concern. In general, the inclusion threshold of 50 years of age needs explanation, as the most liberal definition of older subjects - the UN one - describes them as being 60+. Also, the present cohort 50-69, accounting for well over half of the studied sample, must be split into at least three ranges: <60, 60-64 and 65-69 as these groups will most likely contribute to a better understanding of the phenomena observed. For geriatric care, a differentiation starting with no more than 65 years of age is crucial for understanding the studied parameters and underlying factors. The results thus need to be recalculated.

Line 224 should be removed.

Author Response

Thank you for reading our manuscript carefully. 

We appreciate you for your helpful suggestions. We have added some new findings according to your advices and have accordingly revised our manuscript. We have provided point-by-point responses to your comments.

  • In the context of the reviewed study, the selection of analyzed parameters needs justification. Why is, for example, no parameter of functional capacity included? The conclusions in the manuscript are weak and, as such do not deliver a valid clue.

Thank you for your opinion. We have added functional capacity about speaking ability of Thai language for the ANOVA (page 5, table 2).  In addition, we have added some opinions in the discussion section (page 7, line 217-223).

  • Furthermore, the choice of analyzed age cohorts raises concern. In general,  the inclusion threshold of 50 years of age needs explanation, as the most liberal definition of older subjects - the UN one - describes them as being 60+. Also, the present cohort 50-69, accounting for well over half of the studied sample, must be split into at least three ranges: <60, 60-64 and 65-69 as these groups will most likely contribute to a better understanding of the phenomena observed. For geriatric care, a differentiation starting with no more than 65 years of age is crucial for understanding the studied parameters and underlying factors. The results thus need to be recalculated.

Thank you very much your suggestions. We checked the age of under 60 years old, then just one person was found, so we excluded this person and re-analysed again. Therefore the results were changed many places (such as numbers in tables, and results section). And we have also added new category in the age groups according to your suggestion (page 4, table 1). Then we have re-analysed and added new results in table 2 and 3 (page 5,6, tables 2 and 3). 

  • Line 224 should be removed.

Thanks to your suggestion. We removed "5. Patents" (page 8, line 240)

Round 2

Reviewer 3 Report

The manuscript got substantially better after the review round.